# Healthcare-Associated Adverse Events in Alternate Level of Care Patients Awaiting Long-Term Care in Hospital

**DOI:** 10.3390/geriatrics7040081

**Published:** 2022-08-08

**Authors:** Guillaume J. Lim Fat, Aquila Gopaul, A. Demetri Pananos, Mary-Margaret Taabazuing

**Affiliations:** 1Division of Geriatric Medicine, Department of Medicine, Schulich School of Medicine, Western University, London, ON N6A 3K7, Canada; 2Division of Geriatric Medicine, Department of Medicine, Temerty Faculty of Medicine, University of Toronto, Toronto, ON M5S 1A8, Canada; 3Department of Epidemiology and Biostatistics, Western University, London, ON N6A 3K7, Canada

**Keywords:** delayed discharge, waiting for long-term care (LTC), healthcare-associated adverse events, hospital-acquired infections, healthcare-associated infections, delirium, falls, antimicrobial stewardship

## Abstract

Introduction: A growing number of Canadian older adults are designated alternate level of care (ALC) and await placement into long-term care (LTC) while admitted to hospital. This creates infrastructural challenges by using resources allocated for acute care during disproportionately long hospital stays. For ALC patients, hospital environments maladapted to their needs impart risk of healthcare-associated adverse events. Methods: In this retrospective descriptive study, we examined healthcare-associated adverse events in 156 ALC patients, 65 years old and older, awaiting long-term care while admitted to two hospitals in London, Ontario in 2015–2018. We recorded incidence of infections and antimicrobial days prescribed. We recorded incidence of non-infectious adverse events including delirium, falls, venothrombotic events, and pressure ulcers. We used a restricted cubic spline model to characterize adverse events as a function of length of stay. Results: Patients waited an average of 56 ALC days (ranging from 6 to 333 days) before LTC placement, with seven deaths occurring prior to placement. We recorded 362 total adverse events accrued over 8668 ALC days: 94 infections and 268 non-infectious adverse events. The most common hospital-acquired infections were urinary-tract infections and respiratory infections. The most common non-infectious adverse events were delirium and falls. A total of 620 antimicrobial days were prescribed for infections. Conclusions: ALC patients incur a meaningful and predictable number of adverse events during their stay in acute care. The incidence of these adverse events should be used to educate stakeholders on risks of ALC stay and to advocate for strategies to minimize ALC days.

## 1. Introduction

With the rising care needs of Canada’s aging population, there is corresponding demand for placement of older adults into long-term care centres (LTC). Given limited availability of LTC beds with limited economic investment in opening and staffing new centres, patients often await LTC placement for an extended period following the initial application process, with an average wait time of 159 days for a community application and 90 days for an application from hospital in Ontario [1]. Although the ideal pathway for this transition is for the patient to await placement at home, the need for higher level of care often arises after deterioration of health in hospital. Following an acute illness, older adults often attain a lower functional baseline [2], and when service needs exceed available homecare services, the multimorbid patient is unable to safely return home [3]. Finding themselves with no alternative disposition where they can await placement, a growing number of patients remain in hospital awaiting LTC [4,5].

In Canada, this population of patients are designated “Alternate Level of Care” (ALC) to emphasize their lack of acute medical issues in contrast to the usual hospital patient [4]. In other countries, including the United States and England, these patients are often classified under the “delayed-discharge” designation. In 2008, 5% of all hospitalizations and 14% of all hospital days in Canada were accounted for by ALC-designated patients, and this has continued to grow [4,5]. With particularly long lengths of stay, ALC patients awaiting LTC contribute to a large proportion of non-medical days at acute hospitals [6], creating infrastructural challenges within already strained hospital systems by occupying beds and using resources allocated for acute care. Patient characteristics that have been associated with greater ALC lengths of stay include psychiatric diagnoses such as dementia, behavioral symptoms, cerebrovascular disease, and morbid obesity [6]. This suggests a particular profile of frail, multimorbid, and often cognitively impaired Canadians with challenging care needs making up a disproportionate number of hospital days.

Furthermore, at the patient level, ALC patients are at high risk for individual adverse outcomes. The acute care hospital setting is not designed to meet a patient’s rehabilitative needs but has conversely been shown to advance functional deterioration and place patients at significant risk of hospital-related adverse events including infections and falls [7]. Compared to non-ALC patients, ALC patients have been observed to have longer length of stay, higher median hospital costs, and greater number of complications in hospital, particularly nosocomial infections [7]. At one Canadian academic medical centre, ALC patients were observed to have a median length of stay of 30.85 days, versus 3.95 days in non-ALC patients, and a median hospital cost of $22,459, versus $5003 in non-ALC patients [7]. ALC patients and their families also consistently describe poor care in qualitative studies, with anxiety regarding the uncertainty of the patient disposition and a perception of inconsistent care delivery [8,9].

Our study aims to examine the burden of healthcare-associated adverse events including nosocomial infections, delirium, and falls, in the ALC population and characterize the relationship between these outcomes and length of stay. A secondary aim is to use incidence of these adverse events to educate stakeholders on risks of ALC stay and to advocate for strategies to minimize ALC days

## 2. Methods

In this descriptive retrospective study, we examined the rates of healthcare-associated adverse events in 156 of the 2386 ALC patients who were awaiting long-term care placement while admitted to two acute care hospitals in London, Ontario. The study was approved by the Western University and Lawson Research Institute ethic boards.

### 2.1. Inclusion/Exclusion Criteria

Patients were eligible for inclusion if they were 65 years of age or older, admitted as an inpatient at one of the two hospital sites, did not originally come from a LTC centre, and had been given ALC designation specifically to await LTC placement while all acute presenting issues were resolve.

Patients were excluded if they were younger than 65, came from a LTC centre, or were given ALC designation to await a destination other than LTC (i.e., rehabilitation centre, complex continuing care, or psychiatric facility). We only recorded adverse events during ALC days, thus if a patient became medically active and lost ALC designation, we did not record events until and unless they were designated ALC once more. 

Patients were not excluded if they died while waiting for LTC in hospital.

### 2.2. Sample Selection

We reviewed patient data from University Hospital and Victoria Hospital, two tertiary care centres in London, Ontario, and identified 2386 ALC-designated patients who awaited LTC placement in hospital during 2015–2018. Using a random number generator, a sample of 165 charts were selected. 

The sample size was based on a simulated test of proportions comparing adverse events in ALC patients to patients already placed in LTC, to power detection of a minimum meaningful difference between the two groups of 0.5. While this direct comparative analysis was not performed in this study, ongoing data collection on LTC patients is underway for this secondary study. The simulation recommended 330 patients in total with 165 in each group. We therefore began with a random sample of 165 ALC patient charts, of which 9 were excluded due to being ALC but not awaiting LTC, leaving us with our final sample of 156. Common geriatric syndromes and comorbidities were collected for baseline characteristics.

### 2.3. Data Collection

Charts were individually reviewed by three reviewers using a data extraction tool specifically designed for this study to record the number and types of healthcare-associated adverse events during the entire study period. Active medical issues prior to the date of ALC designation were not included in the recorded adverse events. Healthcare-associated adverse events were classified into two main categories: infectious or non-infectious.

Infectious adverse events were defined as per the McGeer’s Criteria Surveillance Definitions of Infections in Long-Term Care Facilities [10]. These are divided into five groups: respiratory tract infections, urinary tract infections, skin/soft tissue infections, gastrointestinal infections, and bloodstream infections. Each of these groups were further categorized into specific infections, such as pneumonia under respiratory tract infections and *C. difficile* colitis under gastrointestinal infections, as fully outlined in the results. In addition to the different infectious events, we also recorded the number of days of antimicrobial treatment that were prescribed to treat these infections.

We recorded four main non-infectious adverse events of interest: delirium, falls, venothrombotic events (VTE), and pressure ulcers. As above, non-infectious adverse events were only recorded if they were newly developed during the ALC period while awaiting LTC placement. A fifth miscellaneous category of “Other” non-infectious adverse event was also included with a defined selective criterion—such events must be deemed reasonably partially attributable to the hospital environment and/or its associated care and required consensus between two or more of the researchers to be included. Examples included hypervolemia from excessive intravenous fluids, adverse medication reactions, new or worsening depression requiring initiation of antidepressants, worsening anemia in the context of frequent blood draws, and injury from use of physical restraints. 

To enhance consistency between chart reviewers, practice charts were chosen at random from the study population and all three reviewers appraised the same chart independently. Reviewers then compared their individual chart review tools and ensured the same events were recognized as adverse events. This was repeated with three charts. Any discrepancies were discussed until consensus criteria were reached. Formal chart reviewing only commenced following this exercise. 

### 2.4. Statistical Modelling

After the data were collected, we use a restricted cubic spline to model the expected number of adverse events as a function of length of stay in ALC. The spline used four knots placed at the 5th, 35th, 65th, and 95th percentiles of length of ALC stay (7.7 days, 21.9 days, 47.2 days, and 189.1 days respectively). All models were fit using R [11] and the *rms* package [12]. As our model’s primary objective is descriptive, we forgo measures of statistical significance (e.g., *p* values).

## 3. Results

Our study population was a sample of 156 patients of 2386 ALC-designated patients who were awaiting LTC at LHSC between 2015 and 2018. The average age was 84 years. Males made up 51.9% and 46.2% of patients were from home with partner and/or family prior to admission to hospital. The most common comorbidity was dementia, with 46.8% of ALC patients having a documented diagnosis at the time of admission. Table 1 summarizes baseline characteristic of our study population.

Patients waited an average of 56 ALC days before LTC placement, ranging from a minimum of 6 to a maximum of 333 days, with 7 deaths occurring prior to placement. Table 2 shows the different lengths of stay.

For our primary outcome, we recorded 362 total adverse events accrued over the combined 8668 ALC days. Of those, 94 adverse events were infections and 268 were non-infectious adverse events. Table 3 shows all adverse events recorded.

The most common infectious adverse events were urinary tract infections (50 events, 13.81%) and respiratory infections (26 events, 7.18%). The most common non-infectious adverse events were delirium (76 events, 21.0%) and falls (39 events, 10.77%). Non-infectious adverse events included a large proportion of “Other” adverse events which met the aforementioned criteria without fitting into an alternative category, with a total of 129 such events.

The restricted cubic spline model we utilize allows us to non-linearly model expected number of adverse events in our population as a function of the length of stay in ALC. Table 4 shows model estimates for the average number of adverse events as a function of length of stay, using examples of lengths of stay at 14, 30, 60, and 100 days.

Figure 1 shows the plots of model predictions (red) along with data used to fit the model (black circles). To avoid overlapping of data points, we add noise to the vertical component of the scatter plot.

## 4. Discussion

Our results demonstrate that ALC patients incur a significant burden of both infectious and non-infectious adverse events while waiting for LTC, leading to worse patient outcomes, poor antimicrobial stewardship, and further delayed discharges. Our modelling suggests there is a predictability of adverse events in relation to length of ALC stay, which could be used to educate patients and families regarding risks associated with waiting for LTC in hospital. At a systems level, this can also be used to advocate to stakeholders of healthcare administration and hospital leadership to further strategies for reduction of ALC days, appropriate resource allocation, and policy reform.

Our sample reflected the older age of ALC patients, with an average of 84 years, and minimal gender difference with 81 males and 75 females. Previous population-level data obtained using Ontario’s RAI-HC database described a similar average age of 83 years, but slight female predominance at 61.5% female to 39.45% males [3], which we did not re-demonstrate. Unsurprisingly, dementia was our most prevalent co-morbidity at 46.8%. Cognitive impairment has a well-described association with ALC designation, increased care needs, delayed discharge [13,14,15], particularly in Ontario where up to 68.4% of ALC patients have some level of clinical memory impairment [3]. Our random sample was therefore largely representative of previously described characteristics of ALC patients in Ontario.

Hospital-acquired infections (HAIs) are well-described contributors to increased length of stay, morbidity, and mortality at the patient level [16,17,18], while producing a large economic burden at the population level [19,20]. The most prevalent HAIs we observed were urinary tract infections and respiratory infections, and these have been identified as the most common HAIs in older adults [21]. Hospital acquired UTIs are on the rise in Canadian hospitals [22], and a probable contributor is the overdiagnosis and overtreatment of UTIs in hospitalized older adults [23,24]. Hospital-acquired respiratory infections have also interestingly been observed to be over-diagnosed in older adults [25]. A total 620 days of antimicrobial treatment was prescribed for HAIs in our population. While inevitable when treatment is required, increasing overuse of antimicrobials is widely recognized as contributing to growing antimicrobial resistance, particularly in the hospital environment [26,27].

Falls and delirium were our leading non-infectious adverse events, in keeping with their known prevalence and overlap in hospitalized older adults [28]. The fluctuating and often prolonged nature of delirium made recording incidence of delirium distinct, in that we rarely identified multiple convincingly discrete occurrences over the same hospitalization. Delirium was therefore largely binary, either present or absent throughout the hospitalization. The predominance of delirium was reflective of dementia, widely recognized as a strong predisposing risk factor for delirium, being the most common comorbidity in our ALC population at 46.8% of the study population [29].

The ALC designation has several implications that transcends the often-emphasized impact on patient flow across the acute care health system. As defined by the Institute of Medicine, a high-quality health system is safe, effective, patient centered, timely, efficient, and equitable [30]. ALC designation disproportionately impacts older adults with functional impairment and multiple comorbidities including cognitive impairment [8]. We therefore argue that ALC status is an indication of system failure in care quality and equity, placing vulnerable older adults at further risk of functional decline delirium, falls, and infections, while incurring disproportionate healthcare costs [3,31,32]. Our findings reinforce the negative health outcomes detrimental to the individual ALC patient, with an incremental effect with length of stay [32]. Although we did not examine cost, a retrospective cohort study of patients admitted to a tertiary setting similar to our sites confirmed increased adjusted healthcare cost among ALC patients compared to non-ALC patients [7].

Qualitative studies have highlighted the dehumanizing aspect of being an ALC patient, including depersonalization and the notion of “patient over person” while in hospital [8]. ALC patients and their families eagerly await transition into LTC, where they foresee experiencing enhanced autonomy and daily structure [8,9]. We hypothesize that these perceptions and the adverse events we observed stem in part from ALC patients having needs overlooked in favour of patients with acute issues. Furthermore, ALC patients experience multiple relocations as they move through the hospital system. Lack of stability and myriad of unfamiliar environments sensibly increase delirium risk. Given the multiple adverse outcomes described, it is important to advocate for measures and policy reforms which address the overgrowing ALC population and its effect on our strained healthcare system.

### 4.1. Limitations

Our study has a number of limitations. Our data rely on the documentation of these adverse outcomes during admission, and the accuracy and reliability of this charting inherently varies based on physician and allied worker’s practices. Our population is also limited to two specific tertiary hospitals within the same catchment area in the Middlesex–London area, reducing generalizability to other institutions that may have different LTC availability. Our models of adverse events are also descriptive in nature and are limited to predictions within our own study population. While we present our findings stratified within clinically relevant subgroups of interest, namely male/female and dementia/no dementia, our study is not powered to directly compare prevalence of adverse events between these groups or inference of statistical significance, but rather shows descriptive observational data.

### 4.2. Future Directions

Direct comparison to a LTC cohort should be explored to examine if they experience similar rates of adverse events. We are currently pursuing this with data collection at a LTC centre in London, Ontario, with similar catchment area to the hospitals examined.

In recent years, health systems have developed transitional care units (TCU) with the purpose of transferring ALC patients out of acute care beds and into dedicated space more suitable to their level of care. This has been demonstrated to result in improved outcomes for ALC patients at reduced cost [33]. TCU access remains limited, however, and while these can offload a number of ALC patients from hospitals, many remain in acute care settings despite this strategy. Comparing adverse events in a similar TCU population to our cohort would provide further insight into their efficacy.

While operational changes and policy reform are expected, it is yet unclear exactly how the COVID-19 pandemic has affected wait times for LTC and the ALC population [34]. With the disproportionate number of LTC cases and death in Canada [35], it is expected that LTC accessibility has shifted by virtue of both direct resident deaths and changing public perception of safety in LTC. While we expect some degree of risk of adverse events to be specific to the hospital environment, shortcomings in infection control in LTC highlighted by the pandemic raises the question of whether LTC residents truly incur fewer adverse events compared to ALC patients.

## 5. Conclusions

ALC patients incur adverse events while waiting for LTC in an acute care environment maladapted for their needs. This results in a number of downstream effects in an already vulnerable population, disfavored by the limitations of our healthcare system and unfairly perceived as a burden due to associated care costs and bed strain. The predictability of adverse events in relation to length of ALC stay should be used to educate patients and families regarding risk of waiting for LTC in hospital. At a systems level, prevalence of adverse events in ALC patients should be used to advocate for improved homecare resources to support patients at home and solutions to improve access to LTC to minimize waiting in hospital, such as the use of TCUs. Direct comparison to adverse events in LTC and TCUs are avenues for further research.

## Figures and Tables

**Figure 1 geriatrics-07-00081-f001:**
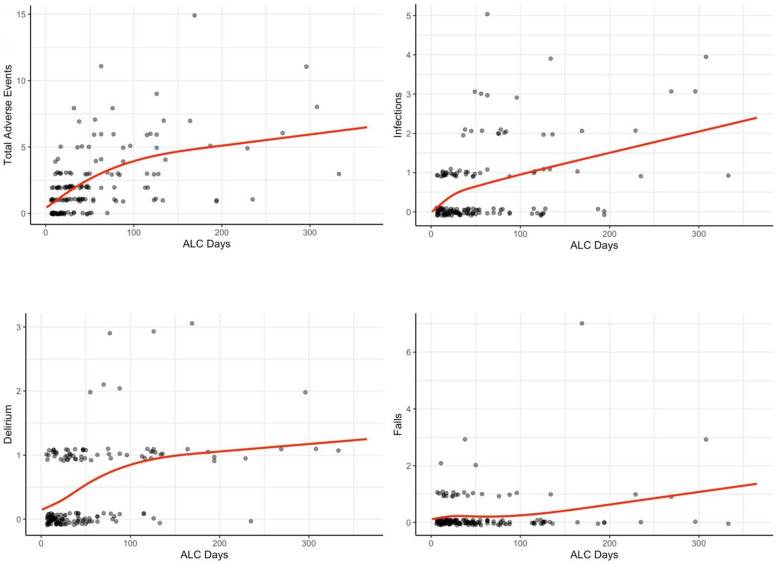
Descriptive modelling of expected healthcare-associated adverse events in study population as a function of ALC days using restricted cubic spline.

**Table 1 geriatrics-07-00081-t001:** Baseline Characteristics.

	All(%)	Females	Males	Dementia	No Dementia
Age					
65–74	26 (16.7%)	12 (16.0%)	14 (17.5%)	6 (8.2%)	20 (24.1%)
75–84	46 (29.5%)	20 (26.7%)	26 (32.5%)	28 (38.4%)	18 (21.7%)
85–94	72 (46.2%)	37 (49.3%)	34 (42.5%)	34 (46.6%)	37 (44.6%)
95 or older	12 (7.7%)	6 (8.0%)	6 (7.5%)	5 (6.8%)	8 (9.6%)
**Sex**					
Male	81 (51.9%)	--	81 (100%)	30 (41.1%)	45 (54.2%)
Female	75 (48.1%)	75 (100%)	--	43 (58.9%)	38 (44.6%)
**Living situation**					
Retirement home	31 (19.9%)	14 (18.7%)	17 (21.0%)	15 (20.5%)	16 (19.3%)
Home alone	48 (30.7%)	29 (38.7%)	19 (23.5%)	15 (20.5%)	33 (39.8%)
Home with partner or family	72 (46.2%)	30 (40.0%)	42 (51.8%)	41 (56.2%)	31 (37.3%)
Other	5 (3.2%)	2 (2.7%)	3 (3.7%)	2 (2.7%)	3 (3.6%)
**Comorbidities**					
Dementia	73 (46.8%)	30 (40.0%)	43 (53.1%)	73 (100%)	--
with BPSD	16 (10.3%)	6 (8.0%)	10 (13.3%)	16 (21.9%)	--
Falls	49 (31.4%)	26 (34.7%)	23 (28.4%)		
Polypharmacy (>10 medications)	46 (29.5%)	21 (28%)	25 (30.9%)	26 (35.6%)	20 (24.1%)
Osteoarthritis	46 (29.5%)	23 (30.7%)	23 (28.4%)	18 (24.66%)	28 (38.4%)
Atrial fibrillation	38 (24.4%)	22 (29.3%)	16 (19.7%)	16 (21.9%)	22 (26.5%)
Diabetes	37 (23.7%)	13 (17.3%)	24 (29.6%)		
Coronary artery disease	26 (16.7%)	8 (10.7%)	18 (22.2%)	13 (17.8%)	13 (15.7%)
Depression	23 (14.7%)	14 (18.7%)	9 (11.1%)	10 (13.7%(	13 (15.7%)
Congestive heart failure	23 (14.7%)	9 (12.0%)	14 (27.4%)	6 (8.2%)	17 (20.5%)
Chronic kidney disease	20 (12.8 %)	8 (10.7%)	12 (14.8%)	12 (16.4%)	8 (11.0%)
Chronic pain	16 (10.3%)	10 (13.3%)	6 (7.4%)	5 (6.8%)	11 (13.2%)
Urinary incontinence	11 (7.1%)	6 (8%)	5 (6.2%)	4 (5.4%)	7 (8.4%)
Parkinson’s disease	9 (5.8%)	4 (5.3%)	5 (6.2%)	4 (5.4%)	5 (6.0%)
Bowel incontinence	8 (5.1%)	5(6.7%)	3 (3.7%)	4 (5.4%)	4 (4.8%)
COPD	8 (5.1%)	5 (6.7%)	3 (3.7%)	3 (4.1%)	5 (6.0%)
Delirium	5 (3.2%)	4 (5.3%)	1 (1.2%)	3 (4.1%)	2 (2.4%)

**Table 2 geriatrics-07-00081-t002:** ALC lengths of stay.

	All (%)	Females	Males	Dementia	No Dementia
<15 days	34 (21.7)	16 (21.3%)	18 (22.5%)	15 (20.5%)	19 (22.9%)
15–30 days	41 (26.2)	22 (29.3%)	16 (20.0%)	9 (12.3%)	29 (34.9%)
31–60 days	37 (23.7)	21 (28.0%)	18 (22.5%)	22 (30.1%)	18 (21.7%)
61–100 days	18 (11.5)	9 (12.0%)	9 (11.3%)	7 (9.6%)	11 (13.3%)
101–200 days	20 (12.8)	7 (9.3%)	13 (16.3%)	15 (20.5%)	5 (6.0%)
201–300 days	4 (2.5)	0 (0%)	4 (5.0%)	3 (4.1%)	1 (1.2%)
>300 days	2 (1.2)	0 (0%)	2 (2.5%)	2 (2.7%)	0 (0%)

**Table 3 geriatrics-07-00081-t003:** Adverse events during ALC stay.

	All	Females	Males	Dementia	No Dementia
Total adverse events	362	156	206	206	156
Infections	94	37	57	58	36
Urinary tract infections	50	22	28	31	19
Respiratory infections	26	6	20	18	8
Skin/soft tissue infections	14	7	7	8	6
Gastrointestinal infections	3	1	2	0	3
Bacteremia	1	1	0	1	0
Non-infectious adverse events	268	119	149	148	120
Delirium	76	35	41	44	32
Falls	39	18	21	20	19
Venothrombotic events	2	2	0	1	1
Pressure ulcers	22	10	12	8	14
Other	129	54	75	75	54
Antimicrobial days	620	247	373	387	233
For urinary infections	299	133	166	200	99
For respiratory infections	147	39	108	79	68
For skin/soft tissue infections	136	51	85	94	42
For gastrointestinal infections	24	10	14	0	24
For bacteremia	14	14	0	14	0

**Table 4 geriatrics-07-00081-t004:** Estimated adverse events over length of ALC stay.

ALC Days	Adverse Events	Infections	Delirium	Falls
14	1.08	0.25	0.23	0.18
30	1.81	0.49	0.35	0.23
60	2.93	0.71	0.62	0.20
100	3.93	0.99	0.85	0.25

## Data Availability

Data is contained within the article. Additional data information available upon request in accordance with ethics boards.

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
