# Peer review of "Healthcare-Associated Adverse Events in Alternate Level of Care Patients Awaiting Long-Term Care in Hospital"

_geriatrics, 2022, doi:10.3390/geriatrics7040081_

Round 1

Reviewer 1 Report

Thank you for your excellent manuscript on an important topic. Overall, I found it to be well-written and very compelling. The introduction provides an excellent background to the ALC system in Canada. While this differs in some respects from healthcare systems in the US and other countries, there are important similarities as older adults wait for LTC placement. I appreciate your decision to perform a descriptive retrospective study and think this enabled you to clearly present the impact for prolonged stays in an ALC system. Frankly, the infectious and non-infectious complications that you observed are what I would have anticipated in this highly complex group of older adults, most of who suffered with some degree of cognitive impairment. Your discussion was thorough and included important limitations to the study. It would be interesting to repeat a similar study post-COVID to evaluate for changes in outcomes given the additional strain on hospitals and LTC facilities.

Author Response

Thank you for taking the time to review our manuscript, and for your positive feedback.

Reviewer 2 Report

The present retrospective study is based on Canadian experience but reflects a generalized problem worldwide that will increase in the coming years with the demographic change. Due to the interest, and since the problem addressed in this topic falls under the concept of ageism, I strongly suggest that this report is included in the Topic Collection on Ageism where following UN recommendations for the Decade of Healthy Aging, we are gathering together scientific evidence of specific gaps and scenarios that need improvement and strategies/interventions to do so. The conclusions of the present report provide new evidence to warn the severe adverse effects of ALC due to ‘Long term care wait times’ in the patients/family and also in the ‘acute care’ and therefore, it is a must to demand a change in the healthcare policy with regards to system performance.  The method to ensure consistency between chart reviewers was done.

In the following paragraphs, this reviewer will visit the different parts of the report constructively to point at some aspects that still may need the authors' attention.

Title

The title is not informative of the findings but refers to a scenario that is already known: that ALC implies adverse events. I’d strongly suggest thinking about which is the main conclusion and home take message that the present retrospective analysis wants to provide to the clinical/scientific community.

Affiliations
First and second author indicate their new current affiliations. This is not feasible. Affiliations refer to which institution/s offered the scenarios to develop a project. Therefore, if the role of the new Universities was to provide such support in the research performances, they should be included as a second affiliation to get that credit. Otherwise, the researcher's profiles at different scientific webs (Academia, ResearchGate, etc.) are there to get updated the new locations of investigators.  

Introduction

Lines 38-39. The authors refer to the limited availability of LTC beds. Please, discuss if this is an indirect indicator, as the limit is the economic investment in opening a new center and personnel.

Line 41 and so on. References. Please, quote the references as indicated per the journal: In the text, reference numbers should be placed in square brackets [ ]

Lines 50-51. No need for brackets when referring to other countries, but the opposite. A sentence to indicate the label used for this scenario is relevant.

Line 66. Aims. According to the data, some recommendations could be given,and this aim should also be included. That is, not just a portrait of the scenario regarding adverse effects and length of stay but what the data tell us about possible solutions. In fact, the last sentence of the abstract refers to it “The incidence of these adverse events should be used to educate 31 stakeholders on risks of ALC stay and to advocate for strategies to minimize ALC days.” So, it should be included as an objective.

Methods

Line 78. The methods refer to a random number generator to select 156 ALC patients. Then, in the “Appendix A- Sample size rationale” an explanation is given, and 165 ALC patients indicated. This paragraph should be better located in the methods.

Line 80. If the authors prefer to let it in the Appendix, then the (see Appendix A for sample size rationale) should be placed immediately after the corresponding sentence, before ‘Common geriatric syndromes…..

Results

Table 1. Baseline Characteristics. According to the age, half of the sample (53,9%, 85-94 plus + 95 or older) is a ‘geriatric’ population, and data could be analyzed in a stratified manner to investigate specific traits, at least for increased vulnerability to the three main adverse events: infections, delirium, and falls. Similarly, from a gender perspective, data should be stratified per sex/gender, as sex is a variable with a biological value.Most importanly, because as the authors discussed, previous results had a higher proportion of females, and the present work may provide clues for male/female differences in the comorbid profile or the variables studied.

Table 2 and 3. More importantly, concerning comorbidities, mental health disorders/diseases should be analyzed in a segregated manner since the level of dependence of these patients is different than the others, and specific profiles in this patient population are relevant for decision making. That is, to build table 2 with 3 more columns, according to segregation per age (geriatric), sex (females), and mental health comorbidity (dementia). In all the cases, the sample size could allow doing so. The same would apply to Table 3.

Although the authors refer to it in the second paragraph of the discussion, it is important to be able to predict the length of ALC according to basic factors, such as age, sex, and comorbidity. This is because, the ‘composition’ of the patient sample (in terms of comorbidities, mainly) determines the outcomes and needs, and just with different sex, age or (mostly) comorbity ‘composition’ of the sample studied, the studied variables (ALC days, total and specific adverse effects) would lead to different results. This is the most important limitation, added to those already stated in the discussion.

Please, note that if this statistical modeling is not applicable to half of the sample (male/female;  geriatric/not; dementia/not) at least, the tables would provide some clues in this respect, and results would be more informative for other clinical settings and/or different ‘comorbidity composition'

The authors found delirium as an important outcome. To which extent the 46.8% of patient population with a dementia diagnosis contributed to it. This should be discussed.

Table 3. and lines 135-139. The category ‘other’ in non-infectious adverse events gathers 129 cases, which implies they are the source of a higher disease burden. I’d suggest also adding the ratio, since it is guessed that the prevalence will be low per each one of them included in this category.

Figure 1. Scatter plots could be illustrated with symbols/colors to identify different comorbidity subgroups. For instance, square (males) and circles, white (organic disease)/black (mental health or dementia).

Discussion

Line 161-162. Why the sentence “there is a predictability of adverse events in relation to the length of ALC stay, 161 which could be used to educate patients and families regarding risks associated with wait162ing for LTC in hospital.” Only patients and families are targeted. Should not the health system, skate holders, etc be also added in the sentence. Nevertheless, they are the ones who handle the responsibility to change the scenario.

After addressing the limitations, the authors discuss recently implemented or future directions.
It would be important to identify these two parts TCU (lines 223-229) and coming years (lines 230-237) in a subsection of ‘recommendations’ or ‘future directions’ or to put them together with ‘Conclusions and Recommendations’.

Reviewer 3 Report

Thanks for recommending me as a reviewer. In this retrospective descriptive study, authors examined healthcare-associated adverse events in ALC patients, 65-years old and older, awaiting long-term care while admitted to two hospitals in London, Ontario in 2015-2018. In this paper, authors  recorded incidence of infections and antimicrobial days prescribed. Authors  recorded incidence of non-ifectious adverse events including delirium, falls, venothrombotic events, and pressure ulcers. Authors used a restricted cubic spline model to characterize adverse events as a function of length of stay. If authors complete minor revisions, the quality of the study will be improved.

1. The introduction section is well written. If the authors describe the trends of previous studies related to alternate level of care patients awaiting long-term care in hospital in more detail in the introduction section, it can help readers understand.

2. line 69-: "METHODS": Authors should be more specific about inclusion and exclusion criteria for subjects in the METHODS section. Whenever possible, authors need to use subheadings to separate paragraphs.

3. line 249: Appendix A: Appendix A is recommends going to the Methods section.

4. Authors should add study limitations to the discussion section.

Round 2

Reviewer 2 Report

The authors have addressed all the suggested issue, the most important referring to the gender aspects. The title is also more precise. The introduction was been improved with portrait of patients and ALC stay characteristics usually associated with the studied scenario. The secondary aim added is relevant as the study provides scientific data on the severe risks/impact of ALC stay useful for stakeholders and policymakers, that at the final end has a frail old person.
The methods have been also improved and are clear on their limitations.

I sincerely thank them for their work.